# FlexMol: A Flexible Toolkit for Benchmarking Molecular Relational Learning

**Sizhe Liu**[1*], **Jun Xia**[2*†], **Lecheng Zhang**[2*], **Yuchen Liu**[1], **Yue Liu**[3], **Wenjie Du**[4],
**Zhangyang Gao**[2], **Bozhen Hu**[2], **Cheng Tan**[2], **Hongxin Xiang**[5], **Stan Z. Li**[2†]

[1]University of Southern California    [2]Westlake University    [3]National University of Singapore
[4]University of Science and Technology of China    [5]Hunan University
sliu0727@usc.edu
{xiajun, zhanglecheng, stan.zq.li}@westlake.edu.cn

## Abstract

Molecular relational learning (MRL) is crucial for understanding the interaction behaviors between molecular pairs, a critical aspect of drug discovery and development. However, the large feasible model space of MRL poses significant challenges to benchmarking, and existing MRL frameworks face limitations in flexibility and scope. To address these challenges, avoid repetitive coding efforts, and ensure fair comparison of models, we introduce FlexMol, a comprehensive toolkit designed to facilitate the construction and evaluation of diverse model architectures across various datasets and performance metrics. FlexMol offers a robust suite of preset model components, including 16 drug encoders, 13 protein sequence encoders, 9 protein structure encoders, and 7 interaction layers. With its easy-to-use API and flexibility, FlexMol supports the dynamic construction of over 70, 000 distinct combinations of model architectures. Additionally, we provide detailed benchmark results and code examples to demonstrate FlexMol's effectiveness in simplifying and standardizing MRL model development and comparison. FlexMol is open-sourced and available at https://github.com/Steven51516/FlexMol.

## 1 Introduction

Molecular relational learning (MRL) aims to understand the interaction behavior between molecular pairs[23]. Among all interaction types, those involving drugs and proteins are of particular interest due to their significant impact on therapeutic discovery and development. Drug-target interactions (DTIs) play a crucial role in various aspects of drug development, such as virtual screening, drug repurposing, and predicting potential side effects [56]. Protein-protein interactions (PPIs) reveal new potential therapeutic targets by enhancing our understanding of protein structural characteristics and cellular molecular machinery [42, 27]. Drug-drug interactions (DDIs) are vital for understanding the effects of concurrent drug use, which can inform strategies to prevent adverse drug reactions and ensure patient safety [5, 12].

MRL has significantly advanced through the integration of deep learning models and substantial AI-ready datasets [14, 54]. MRL models typically consist of several key components: modules for encoding molecule 1, modules for encoding molecule 2, modules for modeling interactions between molecules, and additional auxiliary modules [38, 22, 60, 30, 16, 46, 59, 55]. The variety of encoding methods and interaction layers available allows for the easy construction of new models by recombining these components. This flexibility, however, results in a substantial model space, posing significant challenges for benchmarking processes.

---

*Equal contribution. Correspondence: {xiajun, stan.zq.li}@westlake.edu.cn

38th Conference on Neural Information Processing Systems (NeurIPS 2024) Track on Datasets and Benchmarks.

To address these challenges and facilitate benchmarking in MRL, DeepPurpose [15] emerged as the the first and, to our knowledge, the only MRL library that enables dynamically built models. It offers a user-friendly API, allowing users to create DTI models with customizable drug and protein encoders. However, DeepPurpose inherits several limitations from early MRL models.

- **Insufficient Input Support:** Early models typically relied on amino acid sequences as protein input data due to the limited availability of protein structure data [22, 60, 30]. Consequently, DeepPurpose supports only protein sequences as input types for proteins. However, advances in structural biology have enabled more models to use protein structures—either experimentally verified or AlphaFold-generated—as inputs to enhance performance [46, 11, 58, 55]. This shift underscores the need for libraries that can handle both sequence and structure data for proteins.

- **Lack of Flexible Architectures:** Early machine learning models for MRL often had rigid and simplistic architectures. For instance, DeepDDI uses a Structural Similarity Profile (SSP) to encode each drug [38], while DeepDTI employs convolutional neural networks (CNNs) to encode both drug and protein sequences [60]. These encoded vectors are then concatenated and fed into a multi-layer perceptron (MLP) predictor. Similarly, while DeepPurpose can dynamically construct models, it remains confined to a dual-encoder plus MLP architecture. Recent advances in MRL have introduced more complex and effective model architectures that surpass the capabilities of DeepPurpose. Many models often employ multiple encoding methods for comprehensive molecular representation. Examples include DataDTA [59], which uses pocket structure information along with protein sequences to encode drug targets, and 3DProtDTA [46], which combines Morgan fingerprints and graph neural networks for drug encoding. Finally, interaction layers have become critical components of modern models. For instance, DrugBan [1] uses a bilinear interaction layer, and MCL-DTI [34] implements bi-directional cross-attention to model molecular relations.

In this paper, we introduce FlexMol, a comprehensive toolkit designed to overcome the limitations of existing MRL frameworks. First, FlexMol includes a robust suite of encoders for various MRL input types, featuring 9 protein structure encoders, 13 protein sequence encoders, and 16 drug encoders. Second, it introduces 7 advanced interaction layers, enabling users to build more sophisticated models for accurately modeling molecular relations. Third, FlexMol employs a highly flexible approach to dynamically building models, removing restrictions on the number of encoders and interaction layers, thus allowing for greater customization and complexity. Fourth, we provide a thorough and robust set of benchmark results and comparisons of FlexMol models tested in DTI, DDI, and PPI settings. These benchmarks demonstrate FlexMol's capability to enable researchers to construct, evaluate, and compare MRL models with minimal efforts. Our findings highlight FlexMol's potential to significantly advance MRL and contribute valuable insights to the field.

## 2 Related Work

Several libraries support machine learning-driven exploration of therapeutics, each offering unique functionalities.

**Cheminformatics and Biomolecular Structure Libraries** RDKit is a widely-used cheminformatics toolkit for molecular manipulation, including fingerprinting, structure manipulation, and visualization [21]. Graphein is a Python library for constructing graph and surface-mesh representations of biomolecular structures and interaction networks, facilitating computational analysis and machine learning [17]. However, these libraries are primarily focused on preprocessing tasks and do not directly benchmark MRL models.

**Deep Learning Frameworks for Biomolecular Modeling** DGL-LifeSci leverages the Deep Graph Library (DGL) and RDKit to support deep learning on graphs in life sciences. It excels in molecular property prediction, reaction prediction, and molecule generation, with well-optimized modules and pretrained models for easy application [24]. Therapeutics Data Commons (TDC) provides a unified platform with 66 AI-ready datasets across 22 learning tasks, facilitating algorithmic and scientific advancements in drug discovery [14]. While these frameworks are highly useful, they primarily provide building blocks for MRL model construction and resources for dataset preparation, rather than directly supporting the benchmarking of MRL models.

**Deep Learning Frameworks Specialized for MRL** DeepPurpose is a user-friendly deep learning library for drug-target interaction prediction. It supports customized model training with 15 compound and protein encoders and over 50 neural architectures, demonstrating state-of-the-art performance on benchmark datasets [15].

All these libraries are valuable for molecular relational learning, but DeepPurpose stands out for its ease of use and specialization in MRL prediction. However, its limitations, such as insufficient input type support and rigid model structures, highlight the need for improvement. To address these gaps, we propose FlexMol to offer a more versatile and comprehensive framework for MRL research.

## 3 FlexMol

### 3.1 Molecular Relational Learning

Inputs for MRL consist of pairs of molecular entities including drugs and proteins[14]. Our library specifically targets drug-drug, protein-protein, and drug-target interactions. Drugs are encoded using SMILES notation, a sequence of tokens representing atoms and bonds. Proteins are represented either as sequences of amino acids or/and as 3D structures in PDB format. The primary objective is to develop a function $f : (X_1, X_2) \rightarrow Y$ that maps pairs of molecular representations $(X_1, X_2)$ to interaction labels $Y$. These labels can be binary ($Y \in \{0, 1\}$) for classification tasks such as binding prediction, continuous ($Y \in \mathbb{R}$) for regression tasks such as predicting binding affinity, or categorical ($Y \in \{1, 2, \ldots, K\}$) for multi-class classification tasks such as identifying interaction types.

### 3.2 Framework

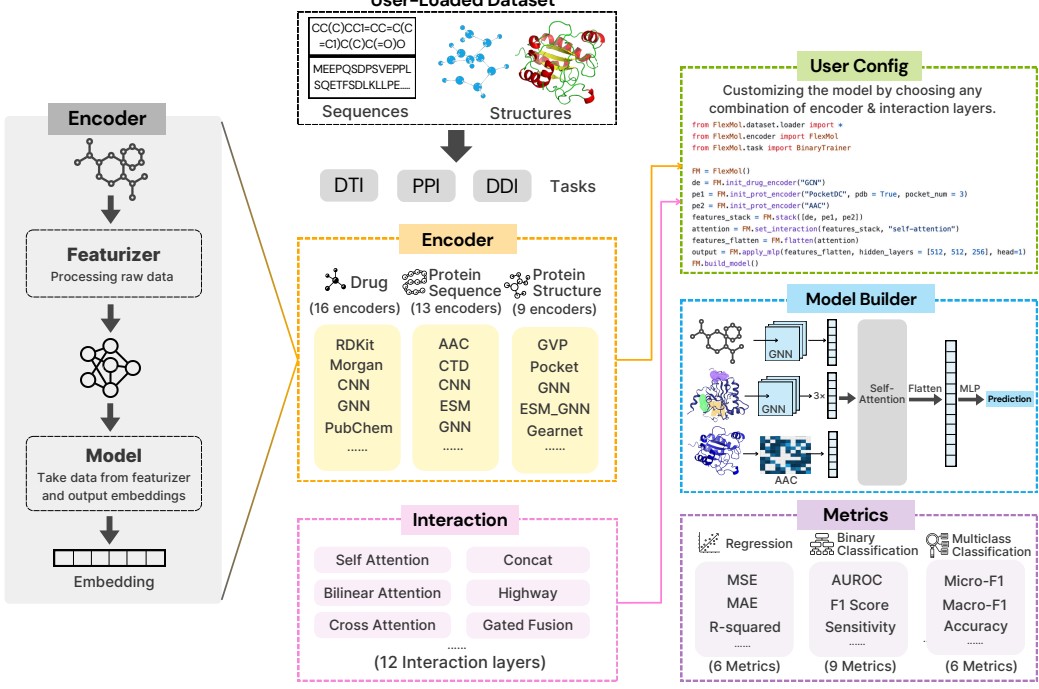

Figure 1: An overview of the FlexMol library.

FlexMol provides a user-friendly and versatile API for the dynamic construction of molecular relation models. Figure 1 illustrates the workflow of FlexMol. First, users select a specific task and load the corresponding dataset. Second, users customize their models by declaring FlexMol components, including `Encoders` and `Interaction Layers`, and defining the relationships between these components. FlexMol offers 16 drug encoders, 13 protein sequence encoders, 9 protein structure encoders, and 7 interaction layers, enabling the construction of a vast array of models. Even without

utilizing interaction layers (defaulting to concatenation) and limiting the total number of encoders to four, FlexMol can generate 71, 368 distinct models for DTI task. Third, once configured, FlexMol automatically constructs the model and manages all aspects of raw data processing and training. The trainer includes presets for a total of 21 metrics for early stopping or testing: 6 for regression, 9 for binary classification, and 6 for multi-class classification.

### 3.3 Components

**Encoder** The `Encoder` class in FlexMol is designed to transform raw molecular data into meaningful representations through a two-stage process: preprocessing and encoding. The preprocessing stage, managed by the `Featurizer` class, involves tasks such as tokenization, normalization, feature extraction, fingerprinting, and graph construction. The encoding stage, handled by the `Encode Layer` class, serves as a building block for dynamically constructing the MRL model. During model training, this class processes the preprocessed data to generate embeddings that are utilized by subsequent layers. Kindly note that for certain fingerprint methods like Morgan[37], Daylight[37], and PubChem[19], which do not have associated encoding layers, a simple multi-layer perceptron (MLP) is employed as the Encode Layer.

An overview of FlexMol encoders is provided in Table 1 and Table 2, with detailed explanations for each method in the *Appendix*. We labeled each encoder with its input type. For drug encoding, "Sequence" refers to the drug sequence, while "Graph 2D" and "Graph 3D" are 2D molecular graph and 3D molecular conformation generated using RDKit from SMILES strings, respectively. For protein encoding, "Sequence" refers to the amino acid sequence, while "Graph 3D" refers to graphs constructed from protein PDB structures.

Table 1: Drug Encoders in FlexMol

| Encoder Type | Methods |
| --- | --- |
| **Sequence** | CNN [15], Transformer [15], Morgan [37], Daylight [37], ErG [41], PubChem [19], ChemBERTa [33], ESPF [15] |
| **Graph 2D** | GCN [20], MPNN [10], GAT [45], NeuralFP [8], AttentiveFP [52], GIN [53] |
| **Graph 3D** | SchNet [39], MGCN [26] |

Table 2: Protein Encoders in FlexMol

| Encoder Type | Methods |
| --- | --- |
| **Sequence** | CNN [15], Transformer [15], AAC [35], ESPF [15], PseudoAAC [4], Quasi-seq [3], Conjoint triad [40], ESM [36], ProtTrans-t5 [9], ProtTrans-bert [9], ProtTrans-albert [9], Auto correlation [13], CTD [7] |
| **Graph 3D** | GCN [20], GAT [45], GIN [53], GCN_ESM [48], GAT_ESM [48], GIN_ESM [48], PocketDC [55], GVP [18], GearNet [57] |

**Interaction Layer** The `Interaction Layer` class is another critical building block of the MRL model. These layers are designed to serve two primary functions: capturing and modeling relationships between different molecular entities, and facilitating feature fusion by combining multiple embeddings of the same entity to create a more comprehensive representation. Interaction Layers can take inputs from various FlexMol components, including Encode Layers or other Interaction Layers, enabling the construction of sophisticated model architectures. FlexMol offers seven preset interaction types for model building: Bilinear Attention[1], Self Attention[44], Cross Attention[34], Highway[59], Gated Fusion[28], Bilinear Fusion[25], and Concatenation, with detailed explanations in *Appendix*.

### 3.4 Evaluation Metrics

The FlexMol Trainer supports multiple default metrics, aligning with the TDC standard for molecular relational learning[14]. Users can specify the metrics in the Trainer for early stopping and testing. These metrics include various regression metrics (Mean Squared Error (MSE), Root-Mean Squared Error (RMSE), Mean Absolute Error (MAE), Coefficient of Determination ($R^2$), Pearson Correlation Coefficient (PCC), Spearman Correlation Coefficient), binary classification metrics (Area Under Receiver Operating Characteristic Curve (ROC-AUC), Area Under the Precision-Recall Curve (PR-AUC), Range LogAUC, Accuracy Metrics, Precision, Recall, F1 Score, Precision at Recall of K, Recall at Precision of K), and multi-class classification metrics (Micro-F1, Micro-Precision, Micro-Recall, Accuracy, Macro-F1, Cohen's Kappa).

### 3.5 Supporting Datasets

FlexMol is compatible with all MRL datasets that conform to our specified format. These datasets typically consist of three components: molecular entity one, molecular entity two, and a label. We provide utility functions to facilitate the loading of datasets in this format. Furthermore, FlexMol includes an interface for loading datasets from the Therapeutics Data Commons (TDC) library, enabling direct loading and splitting of standardized datasets[14]. For more examples and tutorials, please refer to the *Appendix*.

## 4 Experiments

We performed proof-of-concept experiments using FlexMol to show the extensive range of experiments, comparisons, and analyses facilitated by our framework. The following sections present results for DTI experiments, demonstrating the utility of both protein and drug encoders. Additional experiments for DDI and PPI are provided in the *Appendix*.

### 4.1 Experiment #1: Proof-of-Concept Experiments on DTI Task

Table 3: FlexMol Experimental Settings on Davis and Biosnap Datasets

| Experiment No. | Drug Encoder | Protein Encoder | Interaction | Input Feature |
|:---:|:---|:---|:---|:---|
| 1.1 | CNN | CNN | - | $d_s + p_s$ |
| 1.2 | Daylight | CNN | - | $d_s + p_s$ |
| 1.3 | CNN + Daylight | CNN | - | $d_s + p_s$ |
| 1.4 | Morgan | GCN | - | $d_s + p_g$ |
| 1.5 | GCN | GCN | - | $d_g + p_g$ |
| 1.6 | Morgan + GCN | GCN | - | $d_s + d_g + p_g$ |
| 1.7 | CNN | ESM-GCN | - | $d_s + p_g$ |
| 1.8 | ChemBERT | CNN | - | $d_s + p_s$ |
| 1.9 | GCN | ESM-GCN | - | $d_g + p_g$ |
| 1.10 | ChemBERT | GCN | - | $d_s + p_g$ |
| 1.11 | GAT + PubChem | AAC | - | $d_s + d_g + p_s$ |
| 1.12 | GAT + PubChem | AAC | Gated-Fusion | $d_s + d_g + p_s$ |
| 1.13 | Transformer | Transformer | - | $d_s + p_s$ |
| 1.14 | Transformer | Transformer | Cross-Attention | $d_s + p_s$ |

**Note:** $d_s$ = drug sequence, $d_g$ = drug graph, $p_s$ = protein sequence, $p_g$ = protein graph. '-' denotes concatenation for combining embeddings.

Given the extensive number of possible model combinations enabled by FlexMol's flexibility, the goal of this section is not to explore the entire model space. Instead, we use several model combinations as examples to demonstrate FlexMol's robust capabilities in constructing and evaluating diverse model architectures across various datasets and performance metrics.

We utilized the same processed datasets, DAVIS and BIOSNAP, as the MolTrans framework for evaluating DTI[16]. Our setup also integrates AlphaFold2-generated structures to enrich the datasets and enable 3D graph-based protein encoders. Specifically, BIOSNAP includes 4, 510 drugs and 2, 181 protein targets, resulting in 13, 741 DTI pairs from DrugBank [29]. BIOSNAP contains only positive DTI pairs; negative pairs are generated by sampling from unseen pairs, ensuring a balanced dataset with equal positive and negative samples. DAVIS comprises Kd values for interactions among 68 drugs and 379 proteins [6]. Pairs with Kd values below 30 units are considered positive. For balanced training, an equal number of negative DTI pairs are included.

We constructed 14 FlexMol models, detailed in Table 3, and compared them with 8 baseline models (LR[2], DNN[16], GNN-CPI[43], DeepDTI[47], DeepDTA[60], DeepConv-DTI[22], Moltrans[16], 3DProt-DTA[46]). Hyperparameters and codes for each run are available in our public repository.

For both the DAVIS and BIOSNAP datasets, we conducted a random split in the ratio of 7:2:1 for training, validation, and testing, respectively. Each test was repeated five times to mitigate any randomness, and the average results were computed. The experiments were performed using 8 NVIDIA V100 GPUs.

## 4.2 Results and Analysis of Experiment #1

Figure 2 illustrates the example code used to build and run the model for Experiment 1.12. Table 4 presents the performance metrics of the selected baseline models and the FlexMol models, including ROC-AUC (Receiver Operating Characteristic - Area Under the Curve) and PR-AUC (Precision-Recall Area Under the Curve).

```python
# Drug encoder 1: GAT with 64 output features
# Drug encoder 2: PubChem with 64 output dimensions
# Protein encoder: AAC with 64 output dimensions
FM = FlexMol()
de1 = FM.init_drug_encoder("GAT", output_feats=64)
de2 = FM.init_drug_encoder("PubChem", output_dim=64)
pe = FM.init_prot_encoder("AAC", output_dim=64)

# Gated fusion interaction between drug encoders
# Concatenate outputs from drug and protein encoders
de = FM.set_interaction([de1, de2], "gated_fusion", output_dim=64)
dp = FM.cat([de, pe])
dp = FM.apply_mlp(dp, head=1, hidden_layers=[512, 512, 256])

# Prepare datasets and train the model, use roc-auc for early-stopping
FM.build_model()
trainer = BinaryTrainer(
    FM, early_stopping="roc-auc", test_metrics=["roc-auc", "pr-auc"],
    device="cuda:0", epochs=50, patience=10, lr=0.0001, batch_size=64
)
train, val, test = trainer.prepare_datasets(train=train_df, val=val_df, test=
    test_df)
trainer.train(train, val)
trainer.test(test)
```

Figure 2: **Code to Reproduce Experiment 1.12 Using FlexMol.** This example utilizes GAT and PubChem as drug encoders with gated fusion interaction, and AAC as the protein encoder.

**Easy-to-use API**: FlexMol allows for the customization of models in less than 10 lines of code across all 14 experiments. Figure 2 shows the simplicity and efficiency of our API using Experiment 1.12 as an example. Specifically, it takes only 7 lines to customize the model and 9 lines to train and test it.

**Support for Various Input Types**: FlexMol handles diverse molecular data, including drug sequences, protein sequences, and protein structures. Six out of the fourteen experimental combinations use graphs derived from protein structures, demonstrating the framework's strong ability to encode different input types effectively.

**Impact of Additional Encoders:** We utilize Experiment sets {1.1, 1.2, 1.3} and {1.4, 1.5, 1.6} to demonstrate FlexMol's capability to analyze model performance through the integration of additional

Table 4: Performance Metrics on Davis and BIOSNAP Datasets

| Experiment No. / Method | ROC-AUC (Davis) | PR-AUC (Davis) | ROC-AUC (BIOSNAP) | PR-AUC (BIOSNAP) |
|---|---|---|---|---|
| LR[2] | $0.835 \pm 0.010$ | $0.232 \pm 0.023$ | $0.846 \pm 0.004$ | $0.850 \pm 0.011$ |
| DNN[16] | $0.864 \pm 0.009$ | $0.258 \pm 0.024$ | $0.849 \pm 0.003$ | $0.855 \pm 0.010$ |
| GNN-CPI[43] | $0.840 \pm 0.012$ | $0.269 \pm 0.020$ | $0.879 \pm 0.007$ | $0.890 \pm 0.004$ |
| DeepDTI[47] | $0.861 \pm 0.002$ | $0.231 \pm 0.006$ | $0.876 \pm 0.006$ | $0.876 \pm 0.006$ |
| DeepDTA[60] | $0.880 \pm 0.007$ | $0.302 \pm 0.044$ | $0.876 \pm 0.005$ | $0.883 \pm 0.006$ |
| DeepConv-DTI[22] | $0.884 \pm 0.008$ | $0.299 \pm 0.039$ | $0.883 \pm 0.002$ | $0.889 \pm 0.005$ |
| MolTrans[16] | $0.907 \pm 0.002$ | $0.404 \pm 0.016$ | $0.895 \pm 0.002$ | $0.901 \pm 0.004$ |
| 3DProt-DTA[46] | $0.914 \pm 0.005$ | $0.395 \pm 0.007$ | $0.891 \pm 0.004$ | $0.901 \pm 0.014$ |
| 1.1 | $0.894 \pm 0.011$ | $0.347 \pm 0.033$ | $0.873 \pm 0.004$ | $0.874 \pm 0.004$ |
| 1.2 | $0.886 \pm 0.007$ | $0.311 \pm 0.019$ | $0.885 \pm 0.005$ | $0.882 \pm 0.011$ |
| 1.3 | $0.897 \pm 0.008$ | $0.324 \pm 0.009$ | $0.896 \pm 0.003$ | $0.901 \pm 0.004$ |
| 1.4 | $0.890 \pm 0.005$ | $0.330 \pm 0.025$ | $0.872 \pm 0.003$ | $0.879 \pm 0.002$ |
| 1.5 | $0.899 \pm 0.005$ | $0.362 \pm 0.008$ | $0.908 \pm 0.002$ | $\mathbf{0.914 \pm 0.003}$ |
| 1.6 | $0.905 \pm 0.004$ | $0.388 \pm 0.009$ | $0.896 \pm 0.005$ | $0.899 \pm 0.006$ |
| 1.7 | $0.864 \pm 0.009$ | $0.253 \pm 0.018$ | $0.867 \pm 0.002$ | $0.862 \pm 0.004$ |
| 1.8 | $0.902 \pm 0.004$ | $0.380 \pm 0.005$ | $0.893 \pm 0.005$ | $0.891 \pm 0.003$ |
| 1.9 | $\mathbf{0.916 \pm 0.002}$ | $\mathbf{0.408 \pm 0.016}$ | $\mathbf{0.913 \pm 0.001}$ | $0.909 \pm 0.001$ |
| 1.10 | $0.860 \pm 0.008$ | $0.257 \pm 0.010$ | $0.885 \pm 0.009$ | $0.882 \pm 0.019$ |
| 1.11 | $0.884 \pm 0.007$ | $0.295 \pm 0.022$ | $0.900 \pm 0.003$ | $0.899 \pm 0.002$ |
| 1.12 | $0.888 \pm 0.007$ | $0.300 \pm 0.016$ | $0.902 \pm 0.002$ | $0.903 \pm 0.001$ |
| 1.13 | $0.859 \pm 0.005$ | $0.292 \pm 0.013$ | $0.866 \pm 0.004$ | $0.880 \pm 0.005$ |
| 1.14 | $0.902 \pm 0.006$ | $0.376 \pm 0.011$ | $0.893 \pm 0.001$ | $0.905 \pm 0.002$ |

encoders. Specifically, we first examined the effects of combining Encoder A with Encoder C and Encoder B with Encoder C, followed by testing combinations of Encoder A with both Encoder B and Encoder C.

Experiment 1.3, which combined CNN and Daylight for drug encoding with CNN for protein encoding, outperformed both Experiment 1.1 and Experiment 1.2 in both metrics on the BIOSNAP dataset and in ROC-AUC on the DAVIS dataset. Experiment 6, which integrated Morgan and GCN for drug encoding with GCN for protein encoding, showed improvement in both metrics on the DAVIS dataset but a drop in metrics on the BIOSNAP dataset compared to Experiments 1.5.

Our results indicate that integrating additional encoders can improve performance by providing a more thorough representation of the molecule. In Experiment 1.3, the Daylight encoder enriches the CNN encoder by providing chemical information through path-based substructures. In Experiment 1.6, the GCN provides additional graph-based information to complement the sequence-based Morgan encoder. However, this improvement is not guaranteed across all metrics and datasets.

**Impact of Interaction Layers:** We use Experiment sets {1.11, 1.12} and {1.13, 1.14} to illustrate FlexMol's ability to analyze the impact of interaction layers on model performance. Initially, we test the encoder combinations without interaction layers, followed by tests with interaction layers. For example, Experiment 1.12, which employs the Gated-Fusion interaction, outperformed the simpler concatenation method used in Experiment 1.11 across all metrics. Also, advanced interaction layers such as Cross-Attention in Experiment 1.14 further improved model performance compared to Experiment 1.13.

These results indicate that incorporating interaction layers can improve performance when used effectively. The gated-fusion layer in Experiment 1.12 facilitates feature fusion of sequence and graph-level drug representations, while the cross-attention mechanism in Experiment 1.14 enhances the modeling of interactions between substructures of drugs and targets.

### 4.3 Experiment #2: Custom Model Design and Evaluation

This experiment serves as a case study to demonstrate how users can leverage FlexMol to design and construct more complex models.

Figure 3: An overview of the custom model designed using FlexMol.

Figure 3 provides an overview of the new model design. This model builds upon the best-performing model from Experiment 1.9, which utilized a GCN encoder for drugs and a GCN-ESM encoder for proteins. We enhanced this model by adding an additional encoder, "PocketDC," for proteins and incorporating a self-attention module.

To simulate a scenario where users might want to use preset encoders alongside their own custom designs, we reimplemented the GCN as a custom encoder. Then, we built the model by integrating the custom GCN with other preset encoders. The code for defining this custom method and build the whole model is provided in the *Appendix*.

We maintained the same experimental settings as Experiment #1 and conducted ablation studies to simulate real-world scenarios that users might encounter with FlexMol. For the ablation study, "w/o Self Attention" refers to concatenating all outputs from the three encoders. "w/o Pocket Encoder" means removing the additional pocket encoder. "w/o ESM Featurer" involves replacing the GCN_ESM encoder for proteins with a standard GCN.

Table 5: Performance of Custom Model on Davis and BIOSNAP Datasets

| Experiment No. / Method | ROC-AUC (Davis) | PR-AUC (Davis) | ROC-AUC (BIOSNAP) | PR-AUC (BIOSNAP) |
|---|---|---|---|---|
| 1.5 | $0.899 \pm 0.005$ | $0.362 \pm 0.008$ | $0.908 \pm 0.002$ | $0.914 \pm 0.003$ |
| 1.9 | $0.916 \pm 0.002$ | $0.408 \pm 0.016$ | $0.913 \pm 0.001$ | $0.909 \pm 0.001$ |
| Custom Model | $\mathbf{0.926 \pm 0.005}$ | $\mathbf{0.427 \pm 0.014}$ | $\mathbf{0.920 \pm 0.004}$ | $\mathbf{0.919 \pm 0.003}$ |

Table 6: Results of the Ablation Study on DAVIS and BIOSNAP

| Settings | ROC-AUC (DAVIS) | PR-AUC (DAVIS) | ROC-AUC (BIOSNAP) | PR-AUC (BIOSNAP) |
|---|---|---|---|---|
| w/o Self Attention | $0.909 \pm 0.008$ | $0.364 \pm 0.019$ | $0.910 \pm 0.006$ | $0.906 \pm 0.004$ |
| w/o Pocket Encoder | $0.916 \pm 0.002$ | $0.408 \pm 0.016$ | $0.913 \pm 0.001$ | $0.909 \pm 0.001$ |
| w/o ESM Feature | $0.901 \pm 0.006$ | $0.342 \pm 0.013$ | $0.906 \pm 0.005$ | $0.901 \pm 0.002$ |
| Full Model | $\mathbf{0.926 \pm 0.005}$ | $\mathbf{0.427 \pm 0.014}$ | $\mathbf{0.920 \pm 0.004}$ | $\mathbf{0.919 \pm 0.003}$ |

### 4.4 Results and Analysis of Experiment #2

**Extensibility:** This experiment demonstrates FlexMol's ability to easily adapt to user-defined models. By defining encoders according to our protocol, users can incorporate these as building blocks along with other encoders and interaction layers.

**Creating Complex Models:** The custom model involves more complex configurations, including user-defined blocks and additional operations such as stacking and flattening outputs from FlexMol components. Despite this complexity, the model can be constructed in fewer than 10 lines of code after defining the custom encoders.

**Performance:** Table 5 compares the custom model of Experiment #2 with the two best models from the 14 combinations of Experiment #1. Our custom model outperforms all baseline models across all four metrics (ROC-AUC and PR-AUC for both Davis and BIOSNAP datasets).

**Ablation Study:** FlexMol's dynamic model-building capability facilitates easy modification of the model structure for ablation studies. Table 6 shows that the inclusion of additional encoders and interaction layers significantly improves model performance. This improvement is attributed to the additional pocket encoder, which provides atom-level details about potential binding pockets, and the attention layer, which effectively models interactions at the global protein graph level, pocket graph level, and drug graph level.

## 5 Conclusion

We introduced a powerful and flexible toolkit to address the challenges of benchmarking in molecular relational learning. FlexMol enables the construction of a wide array of model architectures, facilitating robust and scalable experimentation. Our framework simplifies the process of model development and standardizes the evaluation of diverse models, ensuring fair and consistent benchmarking.

**Limitations and Future Work**: In the benchmarks presented in this paper, we did not perform an exhaustive combination of model architectures. Our primary goal was to demonstrate FlexMol's implementation and its capability to construct and compare different models. A comprehensive analysis of model combinations was beyond the scope of this paper and is left to the community to explore using our framework. In future work, we plan to continue maintaining and expanding the components implemented in FlexMol. This includes adding more diverse encoders, interaction layers, and evaluation metrics to further enhance the toolkit's flexibility and utility. FlexMol can also be extended to single-instance tasks such as molecular property prediction[49, 50, 51]. Previous research has highlighted the significance of uncertainty prediction in delineating the boundaries of model performance, and how molecular property predictors can serve as feedback to fine-tune generative models[32, 31]. Incorporating these ideas into the FlexMol framework could enhance its effectiveness in benchmarking tasks related to therapeutic discovery.

## Code and Data Availability

The FlexMol toolkit is open-sourced and available on GitHub at https://github.com/Steven51516/FlexMol. The code to reproduce the experiments described in this paper can be found in the `experiments` directory of the repository. The data splits for the DAVIS and BioSNAP datasets were obtained from the MolTrans repository at https://github.com/kexinhuang12345/MolTrans.

## Acknowledgement

This work was supported by National Natural Science Foundation of China Project No. 623B2086 and No. U21A20427, the Science & Technology Innovation 2030 Major Program Project No. 2021ZD0150100, Project No. WU2022A009 from the Center of Synthetic Biology and Integrated Bioengineering of Westlake University, and Project No. WU2023C019 from the Westlake University Industries of the Future Research. Finally, we thank the Westlake University HPC Center for providing part of the computational resources.

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
