# OpenReview forum: "FlexMol: A Flexible Toolkit for Benchmarking Molecular Relational Learning"
_NeurIPS.cc/2024/Datasets_and_Benchmarks_Track — NeurIPS 2024 Track Datasets and Benchmarks Poster_

### Official Review · Reviewer_Bk1c · 2024-07-20
**Review of FlexMol**

**Rating:** 7
**Confidence:** 5
**Correctness:** Yes.
**Clarity:** Yes.

**Review:**

The authors have done an excellent job with the manuscript; the writing is intuitive and makes for a very good read! This reviewer only has suggestions that could further improve the work:

1. When training these models, the authors could mention that uncertainty prediction is a useful metric for understanding regions where property prediction succeeds or fails. A brief discussion of this could be beneficial for readers. The authors could refer to the review paper "Assigning confidence to molecular property prediction." Expert opinion on drug discovery 16.9 (2021): 1009-1023.

2. Another valuable discussion point is that these property predictors can work synergistically with generative models for molecule design. Unlike more expensive and complex methods, these predictors are not limited by the number of evaluations, enabling their use as feedback to further fine-tune generative models. The authors could reference the work "Tartarus: A benchmarking platform for realistic and practical inverse molecular design." Advances in Neural Information Processing Systems 36 (2023): 3263-3306.


Both could be excellent discussion points in the conclusion paragraph.

**Strengths:**

Clearly written and an excellent summary of the challenges/limitations the work is trying to address.

**Additional Feedback:**

None.

**Documentation:**

Yes.

**Ethics:**

No.

**Limitations:**

Yes.

**Opportunities For Improvement:**

All mentioned in the review.

**Relation To Prior Work:**

Yes.

**Summary And Contributions:**

The manuscript presents FlexMol, a toolkit designed to enhance the benchmarking process for Molecular Relational Learning (MRL). MRL is essential for understanding interactions between molecular pairs, which is crucial for drug discovery and development. FlexMol addresses many recent developments in modeling, including advancements in structure prediction, thereby filling gaps left by previous MRL frameworks.

---

> ### Author Rebuttal · Authors · 2024-08-16
>
> Thank you for your constructive feedback and positive comments!
>
> While the primary focus of FlexMol is molecular relational learning, we recognize the potential for extending it to molecular property prediction in the future. We have incorporated a brief discussion of this in the “Conclusion” section. Please let us know if any further modifications are needed. Here is the text we’ve added to the conclusion:
>
> > “FlexMol could be extended to single-instance tasks like molecular property prediction. Previous research has highlighted the significance of uncertainty prediction in delineating the boundaries of model performance, and how molecular property predictors can serve as feedback to fine-tune generative models[1, 2]. Incorporating these ideas into the FlexMol framework could enhance its effectiveness in benchmarking tasks related to therapeutic discovery.”
>
> **References:**
>
> [1] Nigam, A., Pollice, R., Hurley, M. F., Hickman, R. J., Aldeghi, M., Yoshikawa, N., Chithrananda, S., Voelz, V. A., & Aspuru-Guzik, A. (2021a). Assigning confidence to molecular property prediction. *Expert Opinion on Drug Discovery, 16(9), 1009–1023.* [https://doi.org/10.1080/17460441.2021.1925247](https://doi.org/10.1080/17460441.2021.1925247)
>
> [2] Nigam, A., Pollice, R., Tom, G., Jorner, K., Willes, J., Thiede, L. A., Kundaje, A., & Aspuru-Guzik, A. (2023). Tartarus: A benchmarking platform for realistic and practical inverse molecular design. *In Advances in Neural Information Processing Systems* (Vol. 36, pp. 3263-3306).

---

> > ### Comment · Reviewer_Bk1c · 2024-08-26
> >
> > Thank you for the reply & clarification, I also confirm my rating.

---

> > > ### Author Response · Authors · 2024-08-28
> > >
> > > Thank you for your confirmation and for the valuable feedback! Your comments have been very helpful in refining our submission.

---

### Official Review · Reviewer_6jyh · 2024-07-22
**Review for Submission 1106**

**Rating:** 8
**Confidence:** 5
**Correctness:** The reviewer thinks that this paper i…
**Clarity:** This paper is well-written.

**Review:**

**Pros:**
1. FlexMol supports the dynamic construction of MRL models, which aligns with the nature of the problem and enables comprehensive benchmarking across a large model space.
2. The FlexMol API is highly user-friendly, requiring minimal coding effort and lowering the barrier for the general audience.
3. It is the first MRL library to support protein structures in addition to protein sequences and drugs.
4. The paper offers a thorough evaluation of FlexMol baselines, complete with detailed code examples and tutorials to ensure reproducibility.

**Cons:**
1. The paper primarily focuses on benchmarking and does not provide detailed examples of real-world applications, which could help illustrate FlexMol’s practical utility. For example, discussing its application to tasks like drug repurposing would be beneficial.
2. All experiments are performed using a random split. Does the toolkit support other splitting strategies, such as unseen drug/protein splits?
3. The discussion of supported datasets is not detailed enough.

**Strengths:**

1. This work effectively addresses the challenge of managing the extensive model space in molecular relational learning, which is crucial for advancements in drug discovery and development.
2. The toolkit allows the integration of customized encoders alongside preset encoders to build models, providing additional flexibility and adaptability for various research needs.
3. The paper tests several model combinations, offering valuable insights into the effects of additional encoders and interaction layers on model performance.
4. The paper is well-organized and presents ideas clearly. Figures and tables effectively illustrate key points and enhance understanding.

**Additional Feedback:**

1. Provide more insights on how the benchmark can be leveraged to facilitate drug discovery and development.
2. A more detailed documentation of adjustable parameters for each encoder method would improve accessibility .

**Documentation:**

The paper includes a comprehensive appendix with tutorials, code examples, algorithms, and settings.

**Ethics:**

No ethical concerns.

**Limitations:**

The authors discussed the limitations of their work.

**Opportunities For Improvement:**

1. Examples of implementing existing state-of-the-art methods using FlexMol encoder and interaction layer combinations would better demonstrate the robustness of the framework.
2. Does the toolkit support additional new datasets for MRL beyond those available at TDC?
3. Molecular Relational Learning includes several other tasks, such as miRNA-target interactions. Expanding support for these additional tasks would be beneficial for the future development of this project.

**Relation To Prior Work:**

The paper demonstrates clear improvements over prior work (DeepPurpose).

**Summary And Contributions:**

The paper introduces FlexMol, a flexible benchmark for constructing and evaluating diverse model architectures in MRL. This toolkit represents a significant advancement over existing frameworks by enabling the dynamic construction of over 70,000 distinct models and supporting a wide range of data input types. FlexMol simplifies the building of customized models, facilitating the creation of both baselines and new models. Its ease of use makes it accessible to both general audiences and computer scientists. The toolkit’s comprehensive capabilities and support for reproducible research make it a valuable addition to the scientific community.

---

> ### Author Rebuttal · Authors · 2024-08-16
>
> Thank you for your careful review! Below are our detailed responses to your comments:
>
> > Examples of implementing existing state-of-the-art methods using FlexMol encoder and interaction layer combinations would better demonstrate the robustness of the framework.
>
> Thanks for your very valuable advice! We would like to direct your attention to Experiments 1.1 and 1.6, which can be seen as implementations of "DeepDTA" and "3dProtDTA," respectively. Code implementation details can be found in the `experiments/A` directory of our repository. Our results closely match those reported in the original papers, with only minor discrepancies, which can be attributed to the following factors:
>
> 1. We standardized certain layers across all 14 experiments, such as the hidden sizes in MLP layers, which introduced slight deviations from the original implementations.
> 2. To ensure a fair comparison, we standardized the graph construction process for graph-based encoders, such as the node features for proteins, which differ slightly from the original methods.
> 3. Randomness inherent in model training.
>
> > Does the toolkit support additional new datasets for MRL beyond those available at TDC?
>
> Yes, detailed documentation and examples for loading custom data are available in `tutorials/FlexMol_Dataloading.ipynb`. Example custom data files can be found in the `data/toy_data` directory of our repository. Here's a brief summary of the loading process, using DTI data as an example:
>
> - **Option 1:** You can load DTI data using the `load_DTI` function, which supports any file format readable by `pd.read_csv`. Ensure that the first line of your file contains a header with at least the following columns: `Protein`, `Drug`, and `Y`. Optionally, include a `Protein_ID` column if you plan to use a protein structure encoder.
>   - `Drug` is represented by its SMILES string.
>   - `Protein` is represented by its amino acid sequence.
>   - `Protein_ID` corresponds to the PDB file identifier.
> - **Option 2:** Alternatively, you can load the data into a pandas DataFrame using your preferred method without calling `load_DTI`. As long as the DataFrame contains the required columns (`Protein`, `Drug`, `Y`, etc.), you can proceed with the pipeline without any issues.
>
> > Molecular Relational Learning includes several other tasks, such as miRNA-target interactions. Expanding support for these additional tasks would be beneficial for the future development of this project.
>
> Thanks for your very valuable advice! We have currently selected DTI, DDI, and PPI as our primary focus because they are among the most well-known and widely studied interaction types. In the future, we plan to expand our scope to include new tasks like miRNA-target interactions and molecular property prediction.
>
> > All experiments are performed using a random split. Does the toolkit support other splitting strategies, such as unseen drug/protein splits?
>
> Yes, FlexMol does support other splitting strategies, including unseen drug/protein splits. We have demonstrated this capability using FlexMol in conjunction with the TDC library. The detailed tutorial can be found in the file `tutorials/FlexMol_TDC_Interface_Demo.ipynb`. Please note that FlexMol's primary focus is on model building, training, and evaluation, rather than dataset downloading and split preparation. We intentionally avoided duplicating efforts already addressed by established tools to prevent reinventing the wheel.

---

> > ### Comment · Reviewer_6jyh · 2024-08-18
> >
> > Thanks the authors for the rebuttal, which has solved most of my concerns. I recommend acceptance of this paper.

---

> > > ### Author Response · Authors · 2024-08-19
> > > **Thanks very much for the feedback!**
> > >
> > > Dear Reviewer 6jyh,
> > >
> > > Thanks very much for your insightful and helpful reviews, which will undoubtedly help us improve the quality of our article. If our response has successfully addressed your concerns and clarified any ambiguities, we respectfully hope that you consider raising the score. Should you have any further questions or require additional clarification, we would be delighted to engage in further discussion. Once again, we sincerely appreciate your time and effort in reviewing our manuscript. Your feedback has been invaluable in improving our research.
> > >
> > > Best regards,
> > > Authors

---

> > > > ### Comment · Reviewer_6jyh · 2024-08-20
> > > >
> > > > Thank the authors for their rebuttal, which has addressed most of my concerns. I have no further questions and would like to raise the score to 8 in support of this fancy bench.

---

> > > > > ### Author Response · Authors · 2024-08-21
> > > > > **Thanks for Considering Our Rebuttal and Score Adjustment**
> > > > >
> > > > > We are pleased that our rebuttal effectively addressed your concerns. We sincerely appreciate your decision to raise the score and are thankful for your support!

---

### Official Review · Reviewer_yHQP · 2024-07-24
**Review for Paper 1106**

**Rating:** 7
**Confidence:** 3
**Correctness:** Yes.
**Clarity:** Yes.

**Review:**

See below.

**Strengths:**

1. FlexMol provides a robust suite of model components, including 16 drug encoders, 13 protein sequence encoders, 9 protein structure encoders, and 7 interaction layers.
2. It supports dynamic construction of over 70,000 distinct combinations of model architectures, offering flexibility in model development.
3. FlexMol facilitates fair comparison and evaluation of MRL models across different datasets and metrics.
4. The toolkit's easy-to-use API allows for minimal coding efforts to construct and evaluate diverse MRL models.

**Additional Feedback:**

None.

**Documentation:**

Yes.

**Ethics:**

No.

**Limitations:**

See above.

**Opportunities For Improvement:**

Can the benchmark effectively distinguish between small performance differences? If all combinations score similarly, the benchmark design might need improvement.

**Relation To Prior Work:**

Yes.

**Summary And Contributions:**

FlexMol is a comprehensive toolkit designed to address challenges in Molecular Relational Learning (MRL), which involves understanding interactions between molecular pairs, crucial in drug discovery and development. Existing MRL frameworks lack flexibility and scope, leading to repetitive coding and difficulty in fair model comparisons. FlexMol introduces a flexible API supporting dynamic construction of over 70,000 model architectures with various encoders and interaction layers. The toolkit simplifies and standardizes MRL model development and evaluation across multiple datasets and performance metrics, providing benchmark results and code examples to demonstrate its effectiveness.

---

> ### Author Rebuttal · Authors · 2024-08-16
>
> > Can the benchmark effectively distinguish between small performance differences? If all combinations score similarly, the benchmark design might need improvement.
>
> Thank you for your insightful feedback! We agree that while FlexMol supports a broad model space, not all combinations may lead to significant improvements, and some may even offer negligible differences. However, we would like to clarify that the notion that "all combinations score similarly" is a misunderstanding. Here’s our rationale:
>
> (a) While it's true that not every combination leads to major improvements, FlexMol allows us to identify specific combinations that result in statistically significant performance gains. For example, in Experiment 1, where we tested a total of 14 model combinations, the mean metrics across the four metrics (ROC-AUC and PR-AUC for DAVIS and BIOSNAP) were `[0.889, 0.330, 0.889, 0.891]`. However, in Experiment 1.9, where we used the ESM-GCN for proteins and GCN for drugs—a combination not previously explored in recent literature—we observed significantly improved metrics: `[0.916, 0.408, 0.913, 0.909]`. This combination is not only competitive with the mean performance but also with SOTA models. Specifically, it shows a 23.6% improvement in PR-AUC for the DAVIS dataset compared with the mean performance.
>
> (b) Although there's currently no guaranteed method to select the "best model" from the large model space, FlexMol enables us to experiment with different combinations and derive valuable insights. For instance, we observed that an appropriate choice of interaction layer often enhances the modeling of molecular interactions. In our experiments, the inclusion of a "cross_attention" layer consistently enhanced model performance across all metrics in both DTI and DDI settings, indicating that these improvements are unlikely to be due to randomness. These findings are crucial for guiding future MRL model designs.
>
> (c) Beyond benchmarking, FlexMol provides users with the flexibility to design and test custom models by integrating custom layers with preset encoders, interaction layers, and trainers. This capability is particularly valuable for advancing future research, as it allows for rapid construction and evaluation of novel model architectures.
>
> We sincerely hope this clarification addresses your concerns!

---

> > ### Author Response · Authors · 2024-08-28
> >
> > Dear Reviewer yHQP,
> >
> > Thank you once again for your insightful feedback on our work. As we near the end of the discussion phase, we would greatly appreciate it if you could confirm whether our rebuttal has sufficiently addressed your comments and questions, or if there are any remaining areas that require further clarification. We are more than willing to address any remaining concerns.
> >
> > Best regards,
> >
> > Authors

---

> > > ### Comment · Reviewer_yHQP · 2024-08-29
> > > **Official Comment by Reviewer yHQP**
> > >
> > > Thank you for the author’s response. I have raised the score to 7.

---

> > > > ### Author Response · Authors · 2024-08-30
> > > >
> > > > Thank you for acknowledging our response and for the updated score!

---

### Official Review · Reviewer_452B · 2024-07-24
**Review of "FlexMol: A Flexible Toolkit for Benchmarking Molecular Relational Learning"**

**Rating:** 7
**Confidence:** 3
**Clarity:** The article is very well written, suc…

**Review:**

I think this article is very interesting and well written. The FlexMol framework seems to be very flexible, modular, and extensible, and it offers an interesting toolbox to accelerate research in the MRL domain, and to foster reproducibility of research.

**Strengths:**

- Well written article, and interesting
- Clear description
- FlexMol framework advantages are clearly described
- in the limited space of the article the author provide a good description and concrete examples which clarify the use of the proposed framework.
- the framework seems to be very flexible and extensible.

**Additional Feedback:**

No additional comments.

**Correctness:**

The article is clear. I have not tried to use the FlexMol framework, but I think the claims made in the article are correct.

**Documentation:**

The article provides a good and clear overview of the FlexMol framework. The article provide a link to a code repository with further documentation and tutorials.

**Ethics:**

I cannot think of any ethical concern.

**Limitations:**

The paper has a "Limitations" section.

**Opportunities For Improvement:**

Some comment/questions:

1. can the user load custom/proprietary data with FlexMol? How?

2. lines 113-114: "once configured, FlexMol automatically constructs the model and manages all aspects of raw data processing and training." Explain how? data may come in many different formats. See also line 153: "FlexMol is compatible with all MRL datasets that conform to our specified format.": what is your specified format? provide at least link to documentation of the format.

3. can the user add/use custom metrics?

4. suggestion: Table 3: these are "sub-experiments" of Experiment #1 (section 4.1), and therefore I suggest to label them as 1.1, 1.2 ... 1.14 and use these labels also in the text and in Table 5 (1.5, and 1.9 instead of 5 and 9)

5. it would be interesting to see (perhaps in appendix) experiment 1.9 modified with an interaction layer such as cross-attention, to see if that helps in further improving the results.

6. lines 243-244: "Table 5 compares the custom model with the two best models from the 14 combinations". I suggest to write "Table 5 compares the custom model of Experiment #2 with the two best models from the 14 combinations of Experiment #1"

**Relation To Prior Work:**

Prior work is properly discussed.

**Summary And Contributions:**

The article describes FlexMol, a modular toolkit for Molecular Relational Learning (MRL). The authors describe the motivation for their work, and the proposed framework: they provide sufficient high level details about the various components of the framework, and about their intended use. The authors also describe some experiment implemented using FlexMol (which further clarify its usage), and discuss the details and the advantages of using the framework.

---

> ### Author Rebuttal · Authors · 2024-08-16
>
> We appreciate your thorough feedback and the time you took in reviewing our manuscript. Please find our detailed responses to your comments below:
>
> > Can the user load custom/proprietary data with FlexMol? How?
>
> Absolutely! Detailed documentation and examples for loading custom data are available in the `tutorials/FlexMol_Dataloading.ipynb` of our anonymous repository. Example custom data files can be found in the `data/toy_data` directory of our repository. Here's a brief summary of the loading process, using DTI data as an example:
>
> - **Option 1:** You can load Drug-Target Interaction data using the `load_DTI` function, which supports any file format readable by `pd.read_csv`. Ensure that the first line of your file contains a header with at least the following columns: `Protein`, `Drug`, and `Y`. Optionally, include a `Protein_ID` column if you plan to use a protein structure encoder.
>
>   - `Drug` is represented by its SMILES string.
>   - `Protein` is represented by its amino acid sequence.
>   - `Protein_ID` corresponds to the PDB file identifier.
>
> - **Option 2:** Alternatively, you can load the data into a pandas DataFrame using your preferred method without calling `load_DTI`. As long as the DataFrame contains the required columns (`Protein`, `Drug`, `Y`, etc.), you can proceed with the pipeline without any issues.
>
> > Lines 113-114: "Once configured, FlexMol automatically constructs the model and manages all aspects of raw data processing and training." Explain how? Data may come in many different formats. See also line 153: "FlexMol is compatible with all MRL datasets that conform to our specified format.": What is your specified format? Provide at least a link to documentation of the format.
>
> Thanks for your helpful feedback! The specified format is discussed in detail in the previous response. We will also include a link to the relevant documentation in the Appendix.
>
> > Can the user add/use custom metrics?
>
> Yes, the simplest approach is to replace the `trainer.test` function in the regular pipeline with `trainer.inference`. This function returns two lists representing the predicted labels and the ground truth labels, respectively. The user can then analyze the results using any method they prefer, including applying their own custom metrics. An example of using `trainer.inference` can be found in the `tutorials/FlexMol_102_Dual_Encoder.ipynb` file in our repository. We will also reference this in the Appendix.
>
> > Suggestion: Table 3: These are "sub-experiments" of Experiment #1 (section 4.1), and therefore I suggest labeling them as 1.1, 1.2 ... 1.14 and using these labels also in the text and in Table 5 (1.5, and 1.9 instead of 5 and 9).
>
> Thank you for your careful review! We have changed the labels in the revised version.
>
> > It would be interesting to see (perhaps in the appendix) Experiment 1.9 modified with an interaction layer such as cross-attention, to see if that helps in further improving the results.
>
> Thank you for your valuable suggestion! We have explored this idea by modifying Experiment 1.9 to incorporate a cross-attention interaction layer. Since Experiment 1.9 uses two graph-based encoders, setting up cross-attention requires a few adjustments to the original code:
>
> - **(a) Encoder Initialization:** Modify the encoder initialization by setting the parameters `virtual_nodes=True` and `readout=False`. Cross-attention requires 2D tensors as inputs, so the readout layer, which pools all nodes into a global protein representation, is unnecessary. The `virtual_nodes` parameter ensures consistent dimensions for the attention mechanism.
>
> - **(b) Adding Cross-Attention:** Include the following line to apply cross-attention: `att = FM.set_interaction([encoder1, encoder2], "cross_attention")`
>
> Our results indicate that incorporating cross-attention did not lead to significant improvements over the original model when evaluated on the DAVIS dataset. Specifically, the ROC-AUC and PR-AUC scores for the modified model were `[0.917, 0.393]`, which are comparable to the original architecture's scores of `[0.916, 0.408]`. This may be due to the nature of protein sequences; applying attention at the amino acid level might not be optimal. In our dataset, proteins can be up to ~4000 amino acids long, and noise from less relevant regions could overshadow critical interactions.
>
> > Lines 243-244: "Table 5 compares the custom model with the two best models from the 14 combinations". I suggest writing "Table 5 compares the custom model of Experiment #2 with the two best models from the 14 combinations of Experiment #1".
>
> Thank you for your careful review! We agree and have revised the text accordingly.

---

> > ### Comment · Reviewer_452B · 2024-08-20
> > **Response to Rebuttal by Authors**
> >
> > I just want to thank the authors for the useful and interesting rebuttal. I confirm my rating.

---

> > > ### Author Response · Authors · 2024-08-21
> > > **Appreciation for Your Feedback and Confirmation**
> > >
> > > We’re pleased to know that our responses were useful! Thank you again for your constructive feedback throughout this process.

---

### Decision · Program_Chairs · 2024-09-26

**Decision:**

Accept (Poster)

**Comment:**

In the manuscript, the authors describe FlexMol, a toolkit for constructing and evaluating different model architectures for molecular relational learning. FlexMol contains a variety of preset model components, allowing the construction of 70,000 distinct model architectures. The authors also performed a detailed benchmark accordingly.

Strengths:
1. The reviewers found the toolkit useful and highly flexible, and the API easy to use.
2. The reviewers praised the comprehensive comparisons performed in the benchmark.
3. The reviewers found the manuscript well-written, easy to read, and interesting.

Weaknesses:
The reviewers asked some questions about the original version of the manuscript, most of which were subsequently addressed (mostly by clarifications alone without producing extra work) during the rebuttal period. There were a few comments, such as discussing in more detail real-world applications, that could have been addressed more fully.

Overall, all reviewers were positive about the manuscript and consistently recommended its publication.